# Selection and Characterization of Single-Stranded DNA Aptamers of Diagnostic Potential against the Whole Zika Virus

**DOI:** 10.3390/v14091867

**Published:** 2022-08-25

**Authors:** Liliane Monteiro de Morais, Thiago Santos Chaves, Marco Alberto Medeiros, Kaique Alves Brayner Pereira, Patrícia Barbosa Jurgilas, Sheila Maria Barbosa de Lima, Sotiris Missailidis, Ana Maria Bispo de Filippis

**Affiliations:** 1Laboratório de Tecnologia Virológica—LATEV, Instituto de Tecnologia em Imunobiológicos, Fundação Oswaldo Cruz-Fiocruz, Rio de Janeiro 22725, RJ, Brazil; 2Laboratório de Tecnologia Recombinante—LATER, Instituto de Tecnologia em Imunobiológicos, Fundação Oswaldo Cruz-Fiocruz, Rio de Janeiro 22725, RJ, Brazil; 3Vice-Diretoria de Desenvolvimento Tecnológico—VDTEC, Instituto de Tecnologia em Imunobiológicos, Fundação Oswaldo Cruz-Fiocruz, Rio de Janeiro 22725, RJ, Brazil; 4Laboratório de Macromoléculas—LAMAM, Instituto de Tecnologia em Imunobiológicos, Fundação Oswaldo Cruz-Fiocruz, Rio de Janeiro 22725, RJ, Brazil; 5Laboratório de Flavivirus, Instituto Oswaldo Cruz, Fundação Oswaldo Cruz-Fiocruz, Rio de Janeiro 22725, RJ, Brazil

**Keywords:** Zika virus, DNA aptamers, SELEX aptamer technique, viral envelope protein, molecular docking simulation

## Abstract

Zika virus became a major public health problem in early 2015, when cases of Guillain–Barré syndrome and microcephaly were associated with viral infection. Currently, ZIKV is endemic in all tropical areas of the world, and the chance for future Zika epidemics remains very real and accurate diagnosis is crucial. The aim of this work was to select specific ssDNA aptamers that bind to the entire Zika virus and can be used to compose specific diagnostics, without cross-reactivity with other flaviviruses. Zika virus was cultivated in Vero cells and used as a target for aptamer selection. Aptamers specific for the ZIKV were selected using whole-virus SELEX, with counterselection for other flavivirus. Secondary and tertiary structures were evaluated and the molecular anchoring between the aptamers and target were simulated by the HDOCK server. Aptamer interaction was evaluated by ELISA/ELASA and the dissociation constant (Kd) was calculated by thermophoresis. Four ZIKV-specific aptamers were selected. The best two were further characterized and proved to be specific for ZIKV. Aptamers are capable of binding specifically to the ZIKV and differentiate from Dengue virus. The aptamers selected in this work can be used as capture agents in the composition of diagnostic tests to specifically detect ZIKV infection.

## 1. Introduction

Zika virus (ZIKV) is an arthropod-borne virus (arbovirus) primarily transmitted by the Aedes mosquitoes. ZIKV was discovered in 1947 in primates in the Zika Forest of Uganda during surveillance for yellow fever [1,2]. The first documented ZIKV epidemic occurred in the Yap Island in Micronesia in 2007, followed by outbreaks in the Pacific Islands in 2013–2014 [3,4]. In 2015, ZIKV emerged in the Americas, and spread to 87 countries and territories with autochthonous transmission, according to the WHO, by July 2019 [5,6]. Currently, ZIKV is endemic in all tropical areas of the world, similarly to Dengue virus (DENGV), and the chance for future Zika epidemics remains very real [7].

ZIKV is a small enveloped positive-strand RNA virus belonging to the *Flavivirus* genus of the Flaviviridae family. The RNA genome encodes three structural (core C, membrane precursor prM, and envelope E) and seven non-structural genes (NS1, NS2A, NS2B, NS3, NS4A, NS4B, and NS5), with untranslated region (UTR) genes flanking the 5′ and 3′ ends [8,9]. High-resolution ZIKV structures identified by cryo-electron microscopy indicate that the overall ZIKV structure is similar to that of other flaviviruses. The flavivirus E protein, the major protein involved in receptor binding and fusion, is formed as a head-to-tail dimer on the surface of viral particles. There are 90 dimers of the E protein on the surfaces of mature viruses [10]. Each monomer has four domains: the stem/transmembrane domain (E-S/E-TM) that anchors the protein into the membrane and domains I, II, and III (ectodomains) that constitute the predominantly surface portion of the protein [10,11].

Although most ZIKV infections are asymptomatic (75–80%), and symptomatic infections are generally mild [12], the Zika virus became a major public health problem in early 2015, when cases of the Guillain–Barré syndrome, microcephaly, and neurological disorders were associated with virus infection [13,14,15]. Due to its association with congenital infections, ZIKV has been considered the newest member of the TORCH pathogens (toxoplasmosis, others, rubella, cytomegalovirus, and herpes) in the Americas [16,17].

Brain changes caused by congenital Zika appear in the second and third trimesters of pregnancy, while other dysmorphic features are often observed in the newborns of mothers infected with the virus [18,19]. As time is a very important factor in the evolution of Zika infection in pregnant women, accurate and early detection becomes of paramount importance for vertical transmission control, and future therapeutic strategies.

There is an increasing need for affordable, rapid, sensitive, and high throughput methods in identifying new or re-emerging infectious agents [20]. The availability of the point of care (POC) tests eliminates diagnostic delays and uncertainties, enabling the timely initiation of proper treatments and the prevention of the spread of diseases [20]. The use of aptamer-based POC tests has been increasingly described in the literature [21,22].

Aptamers are small single-stranded oligonucleotides simultaneously developed by two groups of researchers, Ellington and Szostak, and Tuerk and Gold, in 1990 [23,24]. They are also called chemical antibodies because of their ability to specifically bind to a target, just like antibodies [25]. However, while antibodies recognize and bind to protein epitope sequences, aptamers recognize and bind based on the 3D structure of the target molecule [26]. In addition to specific, high-affinity binding to their targets, aptamers offer several advantages, including in vitro selection, lack of batch-to-batch variation, room temperature transport, and a relatively low cost, making them ideal for use in diagnostic assays [27].

This paper describes the selection and characterization of specific ssDNA aptamers that bind to the whole Zika virus that can be used to compose a specific diagnostic test without cross reactivity with other Flaviviruses.

## 2. Material and Methods

### 2.1. Zika Virus Strain

The Zika virus isolate ES2916/2015 (accession number: KX212103.1), from the State of Espírito Santo, Brazil, was used as a target in the selection of aptamers.

### 2.2. Other Flavivirus Strains

Other circulating Flaviviruses used were yellow fever virus/YFV 17DD (purified vaccine strain); Dengue 1/WestPac Virus; Dengue Virus 2/S16803; Dengue Virus 3/16,562; and Dengue virus 4/TVP36, with accession numbers DQ100292.1; U88535.1; GU289914.1; KU725665.1; and KC963424.1, respectively. All viruses were supplied by the Laboratory of Virological Technology (Bio-Manguinhos/Fiocruz).

### 2.3. Zika Protein E Gene

The gene encoding the ectodomain of Zika virus E protein (E-ZIKV) was synthesized from the genetic sequence available from GenBank (https://www.ncbi.nlm.nih.gov/genbank/ (accessed on 23 March 2017)—Accession code YP_009227198.1—Envelope protein E (Zika virus)), by the company Integrated DNA Technology (IDT). A polyhistidine tail was inserted into the C-terminal portion to facilitate the further purification of the protein.

### 2.4. Production and Purification of Zika Virus

In order to evaluate the replication profile of the Zika viral isolate, growth kinetics were performed in stationary bottles. This experiment was performed in duplicate, and aliquots were collected daily for the evaluation of viral titer (PFU/mL). Vero cells were seeded at a density of 4 × 10^4^ cells/cm^2^ in 175 cm^2^ flasks 2 days before infection. Virus was inoculated at two different MOIs (0.02 and 0.002) and allowed to bind to cells for 1 h at 37 °C/5% CO_2_. Thereafter, the inoculum was removed, the cells were washed with PBS, and VP/SFM medium supplied with 4 mM glutamine and gentamicin 1% (Thermo Fisher Scientific, Rockford, IL, USA) was added. The virus was harvested every 24 h, the first collection point being eight hours post-infection (hpi). The viral titers were determined by plaque-forming unit (PFU) assay on Vero cells. Briefly, 200 µL of 10-fold serial dilutions were incubated for 1 h on Vero cell monolayers in 6-well plates. The virus inoculum was removed, and the cells were overlaid with medium 199 containing 2% Carboxymethyl cellulose (Sigma-Aldrich, Oakville, ON, Canada), 5% FBS, and 1% penicillin/streptomycin. Four days post-infection, cells were fixed with 5% formaldehyde and stained with 0.04% crystal violet (Sigma-Aldrich) to visualize plaques.

The production of ES 2916/2015 Zika virus was performed using a previously selected viral input and time for harvest. The supernatant was collected and clarified using Sartobran P—0.45/0.2 µm filter capsule (Sartorius). After clarification, the viral suspension was submitted to one purification step of liquid chromatography, in Äkta Purifier 10 Chromatograph (GE Healthcare), using ion exchange chromatography (Sartobind Q75 membrane, Sartorius) in 50 mM Tris/300 mM NaCl/sorbitol 8% solution buffer, pH 8.5. The purified viral titer was calculated using the plaque assay.

### 2.5. Obtaining the Zika Virus Recombinant Protein E (E-ZIKVre)

The E-ZIKVre gene contained in the pUCIDT commercial vector was amplified by PCR with specific oligonucleotides: E-ZIKVre FW (5′-CAT GCC ATG GGC ATT AGG TGC ATA GGC GTT AGC-3′) and E-ZIKVre RV (5′-CCC AAG CTT CTA ATG GTG GTG ATG GTG ATG C-3′) and then cloned into the pET28a expression vector (Novagen, Madison, WI, USA) using restriction endonucleases NcoI (New England Biolabs, Ipswich, MA, USA) and HindIII (New England Biolabs, Ipswich, MA, USA) and the enzyme T4 DNA ligase (Life Technologies, Carlsbad, CA, USA), following the manufacturer’s recommendations. E. coli TOP10 cells were transformed with the pET28a + E-ZIKVre construct by electroporation. Subsequently, the cells were grown on LB agar plates supplemented with 50 µg/mL kanamycin. After overnight incubation at 37 °C, colonies were individually transferred to tubes containing 5 mL of LB medium and 50 μg/mL kanamycin and then incubated for 16 h with shaking at 200 rpm at 37 °C. Plasmid DNA was extracted using the High Pure Plasmid Isolation kit (Roche Life Science, Manheim, Germany). The pET28a + E-ZIKVre purified plasmid was transformed into *SHuffle E. coli* competent cells (New England Biolabs, Ipswich, MA, USA) using electroporation. A single positive clone was picked to inoculate 10 mL of Terrific Broth (TB) medium, supplemented with 50 μg/mL kanamycin and 1% *v*/*v* glucose, and submitted to agitation at 37 °C overnight. Thereafter, 1 mL of bacterial culture was grown in 25 mL of TB medium, containing 50 μg/mL kanamycin and 1% *v/v* glucose, at 37 °C with 200 RPM shaking until obtaining an optical density (OD) of 0.8 at 600 nm. Before induction, a sample of the culture (1 mL) was collected, centrifuged at 14,000 rpm and stored. For protein expression, the culture was induced with 1 mM isopropyl β-D-1-thiogalactopyranoside (IPTG) at 37 °C for 4 h. Then, the OD_600nm_ of the culture was read. The cells were subsequently pelleted by centrifugation and the supernatants were discarded.

### 2.6. Analysis of the Expression and Solubility of the Zika Virus Recombinant Protein E (E-ZIKVre)

The pellet collected in the uninduced culture was resuspended in 25 μL of 1× sample buffer—50 mM Tris-HCl, 2% (*w*/*v*) sodium dodecyl sulfate (SDS), 0.1% (*w*/*v*) blue bromophenol, 10% (*v*/*v*) glycerol, 100 mM 2-mercaptoethanol—for every 0.1 OD_600nm_ and boiled at 100 °C for 5 min.

The induced culture pellets were resuspended in 20 mM Tris-HCl/1 mM EDTA/0,1% triton x-100 buffer and lysed by sonication (3 cycles/30 s) on ice. The insoluble fraction was subsequently separated from the soluble cell lysate by centrifugation (14,000 rpm, 10 min, 4 °C). E-ZIKVre was analyzed using SDS-PAGE and Immunoblot.

### 2.7. E-ZIKVre Protein Solubilization Study

Different formulations of the lysis buffer were used in order to select the best protein solubilization condition. Six buffers containing Tris-HCl, EDTA, triton x-100, and urea at pH 8.0 were tested. The difference between the lysis buffers was only due to the concentration of the chaotropic agent (urea), which ranged from 1 to 6 M. All formulations were added at a rate of 25 μL for each 0.1 OD_600nm_ and subjected to lysis by sonication in 3 cycles of 30 s on ice.

### 2.8. Purification of E-ZIKVre Protein

The culture conditions in the shake flasks were scaled to a 2 L bioreactor for biomass production in order to obtain biomass for the purification step by metal ion affinity chromatography (IMAC). To express the recombinant E Protein at a large scale, the recombinant strain pET28a + E-ZIKVre was inoculated into *E. coli SHuffle* cells grown in a 2 L bioreactor (New Brunswick, St Albans, UK). Biomass growth was induced at an OD_600nm_ of 0.8 with 1 mM IPTG at 37 °C for 4 h. Cell culture was centrifuged at 14,000 rpm for 10 min and the pellet resuspended in wash buffer I [20 mM Tris-HCl + 1 mM EDTA + 0.5% triton x-100 + 1 mM phenylmethylsulphonyl fluoride (PMSF), pH 8.0] in the proportion of 10 mL for each 1 g of pellet. The cell suspension was sonicated for two cycles for 15 s with an interval of 30 s on ice and centrifuged at 8000 rpm for 15 min. The pellet was resuspended in washing buffer II (20 mM Tris-HCl + 1 mM EDTA + 0.5% triton x-100 + 1 M Urea + 1 mM PMSF, pH 8.0) and the processes of sonication and centrifugation were repeated. At the end, the soluble fraction was filtered through a 0.22 μm filter and the recombinant protein was purified by IMAC, using the Ni^2+^ loaded affinity columns of the His Trap HP model (GE Healthcare) in the AKTA Püre equipment (GE Healthcare). Purification took place in steps of increasing imidazole concentration. The purification fractions were analyzed by SDS-PAGE gel and those with the highest concentration were pooled and dialyzed in a step of desalination and refolding of the recombinant protein. The concentration of E-ZIKVre protein was determined by the bicinchoninic acid (BCA) method, using the Pierce BCA Protein Assay kit (Thermo Scientific).

### 2.9. In Vitro Selection of Zika Virus ssDNA Aptamers Using Conventional SELEX

The Systematic Evolution of Binding by Exponential Enrichment (SELEX) procedure was implemented using the protocol described by Simmons et al., 2012, with modifications. A random single-stranded DNA library was used to select sequences that recognize Zika virus particles. This library contained 10^15^ variants of 25 random nucleotides flanked by fixed sequences (5′-GGGAGACAAGAATAAACGCTCAA-25n-TTCGACAGGAGGCTCACAACAGGC-3′). Primers annealing to the flanking regions were used during the selection procedure.

The experiment started with an overnight immobilization of the whole Zika virus to the 96-well ELISA plate at 4 °C. One hundred microliters of the aptamer library (1 mM) were added and incubated for 1 h at room temperature (RT). After incubation, target binding sequences were kept, while unbound nucleotide sequences were removed by washing 3 times with PBS-Tween solution, pH 7.4. The elution of bound aptamers was performed using a salt gradient (1.0 M, 1.2 M, and 1.5 M NaCl) and 3.0 M NaSCN. Eluates were desalted using 5 kDa Microcon filters (Millipore, Life Sciences, Burlington, NJ, USA) and amplified by unidirectional PCR using 10 µL 5× reaction buffer, 1.5 U of Taq polymerase (Invitrogen, Waltham, MA, USA), 200 nM primer sense, 2 nM primer antisense, 20 mM dNTPs, and 50 mM MgCl_2_. Amplification conditions were 10 min at 95 °C; 100 cycles of 90 s at 95 °C; 30 s at 56 °C; 90 s at 72 °C; and a final elongation of 10 min at 72 °C. The SELEX cycle was repeated seven times.

To increase the specificity of aptamers, negative selection was performed with the yellow fever virus (YFV) and the four dengue virus serotypes after the fifth round.

On the last cycle of SELEX, standard PCR of the selected sequences was performed instead, and the quality of the PCR product was verified in 2% agarose gel.

### 2.10. Cloning and Sequencing of Oligonucleotides Enriched by SELEX

The PCR product was cloned into a pCR2.1 TOPO (Promega) vector. The plasmid was transformed into E. coli TOP10 and after electroporation, the LB medium was added and the material was grown for 1 h at 37 °C, 200 RPM. The culture was then plated onto LB/agar/ampicillin (100 μg/mL) and the material was incubated overnight at 37 °C. The viable clones were then individually seeded from the colonies in 5 mL of LB/ampicillin and incubated overnight at 37 °C, 200 RPM. The plasmids were extracted following the manufacturer protocol of the High Pure Plasmid Isolation Kit (Roche, Manheim, Germany). The extracted plasmids were quantified by Qubit (Invitrogen, Waltham, MA, USA) and amplified using M13 forward and M13 reverse primers. Sequencing was performed using Sanger’s automated methodology using the Big Dye kit, version 3.1, protocol and the 3500 XL equipment (Applied Biosystems, Waltham, MA, USA). Selected ssDNA molecules were aligned by Clustal 1.2 (Version 1.2.4, Institute Pasteur, Paris, France).

### 2.11. Generation of Two- and Three-Dimensional Structure

The secondary structure of ssDNA was constructed from their nucleotide sequences using the Mfold Web Server [28]. The sequences were selected as linear at a temperature of 25 °C and ionic concentration of 0.1 M of Na^+^, 0.01 M of Mg^2+,^ computing only fold configurations within 5% from the minimum free energy, and considering a maximum number of 50 folds with no limit to the maximum distance between paired bases. In addition to the predicted secondary structure of ssDNA, the minimum free energy of the predicted structure is provided.

Three-dimensional modeling was performed using the protocol described by JEDDI and SAIZ, with modifications [29]. Briefly, using the RNAalifold Web Server, ssDNA aptamer sequences in FASTA format were converted into ssRNA (Vienna format), containing the coordinates of the three-dimensional structure. The digital platform “RNA Composer” was used to transform structures from ssRNA in Vienna format into ssRNA in PDB format. Equivalent 3D ssRNA models were constructed using the VMD software, and the 3D RNA models were translated into DNA models using the Chimera program. The final 3D ssDNA structures were visualized by the Chimera software.

### 2.12. Molecular Dynamics Simulations between Zika Virus E Protein and Aptamers

The Zika virus E protein [30], available in the Protein Data Bank (PDB) database [31], and the 3D ssDNA structures of the aptamers, finalized in the Chimera software, were applied to simulate molecular dynamics (molecular docking) using the HDOCK platform.

### 2.13. Binding Affinity of Selected Aptamers

#### 2.13.1. ELISA

Enzyme immunoassay plates were coated with 50 µM/well of poly-lysine (PL) at 5 μg/mL then stored at 4 °C for 16 h. Aptamers were immobilized on wells and the plate was incubated overnight at 4 °C. To remove unbound aptamers, wells were washed three times with washing buffer (10 mM PBS, 0.05% Tween-20, pH 7.4,). The plate was blocked with 300 μL/well of a blocking agent (10 mM PBS, 1% BSA, pH 7.4) for 1 h at 37 °C; and another cycle of washes followed and 100 μL/well of different concentrations of virus in 10-fold serial dilutions (5,000,000–0 PFU/mL) was added. After incubation for 1 h at 37 °C, the unbound virus was removed by washing three times. One hundred microliters of monoclonal antibody 4G2 in a 10-fold dilution was added and the plate was incubated at 37 °C for another 1 h. Subsequently, the plate was washed again, 100 µL of anti-mouse conjugated with peroxidase at a dilution of 1/30,000 in PBS was added and the plate was incubated for 1 h at 37 °C. The plate was again washed three times and 100 µL of tetramethylbenzidine (TMB) chromogenic solution was added before 20 min of incubation at 37 °C. The color reaction was stopped by adding 100 µL of 1 N HCl and the absorbance was read at 450 nm using the Spectra Max190 reader (Molecular Devices, San Jose, CA, USA) supplied with the Soft Max Pro 6.3 software.

#### 2.13.2. Affinity Binding by Microscale Thermophoresis

Microscale thermophoresis (MST) was performed on a NanoTemper Monolith NT.115 apparatus (Nano Temper Technologies). Briefly, 10 μM Zika virus E protein was labeled using a Monolith Protein Labeling Kit RED-NHS 2nd Generation, with 3-fold excess NHS dye in PBS (pH 7.4). The free dye was removed according to the manufacturer’s instructions, and the protein was eluted in the MST buffer (PBS containing 0.05% Tween 20) and centrifuged at 14,000× *g* for 10 min. Binding affinity measurements were performed using 20 nM of labeled protein and a serial dilution 1:1 of 16 concentrations of the four aptamers, starting at 50 μM. The protein was maintained in the presence of the aptamers for 15 min prior to transferring to premium Monolith NT.115 capillaries. Experiments were run in duplicate at 76% excitation and a medium MST power at 25 °C. Data were obtained with MO.Control V2.1.1 software (NanoTemper Technologies, Munich, Germany) and recorded data were analyzed with MO.Affinity Analysis V3.0.4 software (NanoTemper Technologies, Munich, Germany). The dissociation constant (Kd) quantifies the equilibrium of the reaction of the labeled molecule A (concentration cA) with its target T (concentration cT) to form the complex AT (concentration cAT) and is defined by the law of mass action as Kd = cAxcT/cAT, where all concentrations are “free” concentrations. During the titration experiments, the concentration of the labeled molecule A is kept constant, and the concentration of added target T is increased.

In order to assess the specificity of aptamers, we used a pool of viral particles containing all four Dengue serotypes. Two aptamers were selected and labeled with the second-generation RED-NHS labeling kit. The reaction was carried out according to the manufacturer’s instructions in buffer 130 mM NaHCO3, 50 mM NaCl, pH 8.3, applying 10 μM of aptamer (in molar ratio to fluorophore ≈ 1:3) at room temperature for 30 min. Unreacted fluorophore was removed using a size exclusion column equilibrated in PBS buffer, 0.05% Tween 20, pH 7.4. Binding affinity analyzes were conducted using labeled aptamers adjusted to 40 nM with PBS buffer, 0.05% Tween 20, 150 mM NaCl, pH 7.4. For measurement, each virus dilution (ZIKV and DENGV pool) was mixed with a volume of labeled aptamer, at a final aptamer concentration of 20 nM and final virus concentrations ranging from 0.00000652 nM to 0.01335 nM. After 15 min of incubation, samples were loaded into standard capillaries. The effect of thermophoresis was measured using the Monolith NT.115 equipment at room temperature of 25 °C. The parameters were set for 100% or 64% excitation (ZIK01/NH2 or ZIK02/NH2 aptamers, respectively) and average LED power. Data from the duplicates were analyzed using the MST signal at 5 s.

## 3. Results

### 3.1. Zika Virus Production

Before creating a working virus seed (WVS), a growth kinetics of the virus in Vero cells was performed to evaluate the replication profile of the Zika virus isolate ES2916/2015. The virus was inoculated at the MOIs 0.02 and 0.002 and was collected every 24 h, the first collection point being 8 hpi. Compared to the cell control, ZIKV-infected cells demonstrated no apparent cytopathic effect (CPE) up to 48 hpi for both MOIs. Infected cells demonstrated mild CPE by 72 hpi, moderate CPE by 96 hpi, and extensive CPE by 96–120 hpi (Figure 1A).

Viral kinetics using MOI 0.02 showed faster growth, as expected, and reached a plateau 12 h before the MOI 0.002 curve. The peak of viral production is between 48 and 72 h at MOI 0.02 (6.19 and 6.21 Log10 PFU/mL, respectively) and 72 h, at MOI 0.002 (6.32 Log10 PFU/mL) (Figure 1B).

For working seed bank production, Vero cell cultures were inoculated with a viral input of 0.02 and the virus was collected after 48 h of infection. Following the purification of the ZIKV, a viral titer of 7.44 Log10 PFU/mL was obtained.

### 3.2. Cloning, Expression, and Purification of E-ZIKVre Recombinant Protein in E. coli SHuffle System

For the expression of the recombinant protein E-ZIKVre, the gene was amplified by PCR. In Figure 2A, it is possible to observe a band of approximately 1247 bp corresponding to the E-ZIKVre gene. This gene was cloned into pET28a vector and E. coli TOP10 cells were transformed with the pET28a+E-ZIKVre construct. Plasmids of possible recombinant clones were treated with restriction endonucleases NcoI and HindIII, where a band corresponding to the E-ZIKVre insert was observed (Figure 2B). Clone 3 was selected to transform *E. coli SHuffle* cells.

The expression and solubility of the E-ZIKVre protein was evaluated in induced and non-induced cell extracts using SDS-PAGE (Figure 3A) and immunoblotting (Figure 3B). The expected molecular mass (45 kDa) was observed both in the samples referring to the induced cells and in the samples of the insoluble fraction in SDS-PAGE. The immunoblotting assay with anti-his antibody confirmed these results.

Protein solubilization assays were performed and analyzed by SDS-PAGE with different concentrations of chaotropic agent (urea). The condition of 20 mM Tris buffer + 1 mM EDTA + 0.5% triton x-100 + 4 M urea pH 8.0, 3 cycles of 30 s and 30 s intervals on ice was established as ideal to obtain the complete solubilization of the E-ZIKVre protein (Figure 4).

Cell culture conditions were scaled-up to a 2 L bioreactor in order to obtain sufficient biomass for protein purification. The E-ZIKVre protein was purified by IMAC (Figure 5A) and eluted in buffer containing 500 mM imidazole. Fractions collected during elution were analyzed by SDS-PAGE (Figure 5B), with a high degree of homogeneity >85%. After the purification step, the E-ZIKVre protein was dialyzed to remove the chaotropic agent and refold. The quantitation results demonstrated a recovery of E-ZIKVre protein of approximately 37 mg/L.

### 3.3. ssDNA Aptamers against Zika Virus

SELEX technology was used for the selection of specific DNA aptamers against whole viral particles. A randomized ssDNA library was incubated with Zika virus particles immobilized on ELISA plates and cycles of interaction, and the separation of the bound from unbound species as well as the elution and amplification of the bound species were performed. After the fifth selection round, negative selection was performed with others flaviviruses, the four dengue serotypes and yellow fever virus, to increase the specificity of aptamers. At the end of the SELEX cycles, the PCR-derived molecules were cloned and sequenced. Ninety-four clones containing an aptamer insert were obtained by elution with 3 M NaSCN. The sequencing of 28 of these clones was performed and their homology was evaluated using a Clustal 2.1 alignment (Figure 6). The formation of four aptamers clusters with 100% nucleotide homology was detected: cluster ZK01 (aptamers ZK04 and ZK83); cluster ZK02 (aptamers ZK06, ZK23 and ZK78); cluster ZK03 (aptamers ZK34 and ZK88); and cluster ZK04 (aptamers ZK17 and ZK86). In addition, the homology of 22 nucleotides (GTTTAGAAACCCCACCGTGGCA) was observed in 10 sequences (Figure 6). The four clusters were chosen from the selected pool for further characterization. Clusters ZK01 and ZK02 were analyzed using their complete sequence, whereas the sequences of clusters ZK3 and ZK04 were truncated to their appropriate predetermined size.

### 3.4. Secondary and Tertiary Structure Prediction of the ssDNA Aptamers

Secondary structures of the ssDNA molecules were predicted using the Mfold Web Server, based on free energy minimization techniques. In mfold, all possible secondary structures are approximated based on Watson–Crick pairing and the most thermodynamically stable structures are selected.

Figure 7A shows the most stable two-dimensional structures of each aptamer group and their respective minimum-fold free energies (ΔG). Clusters ZK02, ZK03, and ZK04 demonstrated a conservation of structure, a hairpin formed by the final 11 nucleotides of the forward primer and the first five nucleotides of the aptamer sequence.

The transformation of secondary sequences into three-dimensional structures consisted of multiple steps and the use of two programs, VMD and chimera. The results of the three-dimensional analysis confirm the structural similarity between aptamers ZK02, ZK03, and ZK04 (Figure 7B).

### 3.5. Molecular Docking between ssDNA Aptamers and the E-ZIKVre Protein

Protein E is a glycoprotein that involves the entire surface of flavivirus viral particles. Each glycoprotein E monomer forms stable homodimers, and 90 dimers assemble to form the outer envelope of the infectious virus. Protein E is the main target of neutralizing antibodies against the Zika virus and the likely target of the selected aptamers. We verified this assumption using recombinant protein E in thermophoretic analyses, to confirm that the aptamers indeed bind to this protein, verify the binding profile, and calculate the dissociation constants. We also performed the assays with the intact virus to confirm the aptamer-target specificity, using a Dengue virus pool as a control (see next section).

Having confirmed the interaction of the selected aptamers with the E protein, molecular dynamics simulation between the aptamers and ZIKV E protein was performed using the HDOCK server. HDOCK performs global docking to predict binding complexes between two molecules using a hybrid docking strategy.

We analyzed the first 10 predictions and eliminated those that interacted in the unexposed part of the protein, facing the transmembrane region of protein E of ZIKV. Among the models linked to the outer surface of the protein, we selected the one with the lowest docking score. For aptamers ZK01 and ZK02, model 1 was selected. The aptamers ZK03 and ZK04 presented model 3 as the model with the lowest energy that met the pre-established parameters. Figure 8 shows the sites of interaction between each aptamer and the dimeric E protein.

It is observed that all aptamers present binding sites on two adjacent monomers of the same E-dimer (Figure 8 and Table 1).

### 3.6. ELISA

An in-house ELISA/ELASA was used as a screening tool to assess the performance of the ZK01, ZK02, ZK03, and ZK04 aptamer clusters. Aptamers were tested against the serial dilution of Zika virus at base 10. Optical density data demonstrated that all sequences were only able to recognize Zika virus at a dilution of 5 × 10^6^ PFU/mL (Figure 9).

### 3.7. Microscale Thermophoresis

Identification and quantification of interactions between ZIKV and aptamers was analyzed by microscale thermophoresis (MST) using a NanoTemper Monolith NT.115 apparatus, aiming to determine the dissociation constants (Kd) and confirm the specificity of the aptamers.

Labeling of the E-ZIKVre protein was performed with reactive dyes using N-Hydroxysuccinimide (NHS)–ester chemistry, which reacts efficiently with the protein’s primary amines to form highly stable dye–protein conjugates. The efficiency of the labeling step was evaluated by preliminary tests available on the equipment, and the intensity of gross fluorescence within the detection limit could be verified. After the preliminary test, the binding check test was performed to assess whether there was a detectable response amplitude between the target (labeled E-ZIKVre protein) and the ligands (aptamers) at the maximum concentration available. In this test, one can select the best assay buffer for the protein, targeting an optimal MST binding assay. With this, it was possible to evaluate the conditions of protein aggregation, adsorption in the capillaries, fluorescence intensity, and homogeneity to obtain a higher signal/noise ratio and establish the assay conditions to carry out the quantification of the affinity through the binding affinity test.

To investigate the binding affinity between E protein and Zika virus aptamers, 20 nM of the labeled protein (target) and serial dilutions (50–1.53 nM) of the four aptamers were used. As shown in Figure 10, the binding affinity of aptamer ZK01 for E-ZIKVre (Kd = 1414.90 ± 0.41 nM) was slightly higher than for the other aptamers with the greater homology (ZK02 Kd = 761.54 ± 0.49 nM; ZK03 K: 919.26 ± 0.68 nM; ZK44 Kd: 820.98 ± 0.34 nM) (Figure 10).

The evaluation of the specificity of the aptamers was verified using a pool of Dengue serotypes in comparison with the Zika virus, as DENGV is well-known to cross-react with Zika in many diagnostic tests. According to Figure 11, the variation in the concentrations of the DENGV pool did not change the normalized fluorescence response, which suggests no interaction in the concentration range evaluated. On the other hand, the variation of ZIKV concentration caused an increase in the response at higher concentrations, which indicates the presence of intermolecular interaction between aptamers and ZIKV (Figure 11). As there was no complete saturation of the curve, due to insufficient viral titters, the dissociation constant (Kd) could not be calculated.

## 4. Discussion

Aptamers have been extensively used in the study of arbovirus with some very interesting results, both for diagnostic and therapeutic applications, which have been recently reviewed by our group [32]. Particularly in the area of diagnostics, they have been capable of separating between different Dengue serotypes, but have also been applied for ZIKV, tick-borne encephalitis virus (TBEV), Rift Valley fever virus (RVFV) and Chikungunya Virus (CHIKV) with considerable success [32]. In the case of previous Zika studies using aptamers, the authors used the NS1 protein as a target, and in most cases, their selected aptamers are not capable of differentiating between ZIKV and DENGV NS1 proteins, with one exception [32]. In this study, we are presenting for the first time aptamers selected against the whole viral particle, verifying its binding to the whole virus and the E protein specifically, and demonstrating the specificity of that aptamer and its capacity to differentiate between zika and dengue viruses.

Several studies in the literature have reported the expression of structural and non-structural flavivirus proteins using the *E. coli* prokaryotic system [33,34,35,36]. This system has the advantage of enabling high levels of expression at a relatively low cost. The flavivirus envelope protein is the most abundant on the surface of the viral particle. Despite the system’s limitation in carrying out post-translational modifications, such as glycosylation, immunogenicity assays using flavivirus envelope proteins produced in the prokaryotic system demonstrated the ability of this protein to induce the formation of neutralizing antibodies [37,38], indicating the conservation of antigenic epitopes.

The recombinant protein E-ZIKVre was expressed in the form of inclusion bodies, as shown in other works in the literature [33,37]. Yields of E-ZIKV protein expression in the prokaryotic system of 0.55 mg/mL and 10 mg/L were reported by Han et al. [37] and Liang et al. [39], respectively. In this work, we obtained a final yield after purification of 37 mg/L of culture at a final concentration of 0.76 mg/mL with a purity > 90%, demonstrating the efficiency of the strategy used.

Although different strategies have been adopted for the selection of aptamers against different arboviruses, as shown above [32], their binding affinity can be influenced by the stringency of the selected conditions, such as, for example, reducing the target concentration or modifying the binding and washing conditions [40]. In this work, we used a protocol based on the interruption of interactions formed between the aptamer and the target using buffers with a high salt content and chaotropic agents. In a previous study carried out by Simmons et al. [41], using the same selection conditions, high homology was observed in the sequences generated in both 1.5 M NaCl and 3 M NaSCN elutions, and the performed assays confirmed binding with the target. In our experiments, in addition to the use of the same elution conditions, we introduced negative rounds of selection against the yellow fever virus and the different dengue serotypes, aiming to select not only the highest binding aptamers, but the ones that were only specific to Zika virus and not for other circulating flaviviruses in the country. We used aptamers resulting only from the 3 M NaSCN elution and we achieved the formation of four aptamer clusters and a highly conserved region of 22 nucleotides and a subset of complete sequence identity, indicating a core sequence necessary for binding, with some variability permitted in the formation of the additional binding loop.

The affinity of the specificity of the aptamers for the ZIKV viral particles was studied by ELISA and microscale thermophoresis, confirming the ability of the aptamers to recognize both the entire viral particle, and the E protein in particular. In the ELISA assay, the negative charge on the DNA backbone, due to the phosphates, makes it difficult for aptamers to adhere to polystyrene plates. A practical and inexpensive alternative is the use of polylysine to modify the surface charge of the bottles and plates for cell culture [42] and plates for enzyme assays [43]. We developed an in-house ELISA with polylysine-coated plates that permitted the immobilization of aptamers in order to perform the ELISA assays. Using this method, we demonstrated that the aptamers were capable of recognizing the ZIKV in ELISA assays. The sensitivity of the assay could potentially be improved, as it is understood that the use of lysine permits de deposition of the aptamers in a disorderly manner, not optimizing the exposure of the aptamer binding sites. Alternative methods of immobilization using biotin–streptavidin systems, or gold nanoparticles with thiol-modified aptamers could improve the performance of the system and permit direct use in immunochromatography-based tests. Nevertheless, our current results permit the verification of the aptamers to bind to whole viral particles. This was further confirmed in the MST, where it was possible to identify not only the interaction of the aptamers to ZIKV, but also the inability of the aptamer to interact with DENGV particles at the same viral count.

Finally, it was supposed that the most probable target on the viral particle surface would have been the E protein. This was verified by MST, where the affinity of the aptamer for a recombinant purified ZIKV E protein was calculated. Post-SELEX modifications with functional groups have been described to optimize the detection (fluorescent molecules) or immobilization (biotin, amine) of aptamers [40]. In the development of this project into a viable diagnostic, we aimed to investigate post-SELEX modification to increase aptamer affinity, as well as optimize the assay development format to take advantage of the capability of the aptamer to bind to ZIKV and differentiate from DENGV particles, into developing novel diagnostic strategies for Zika virus infections.

## Figures and Tables

**Figure 1 viruses-14-01867-f001:**
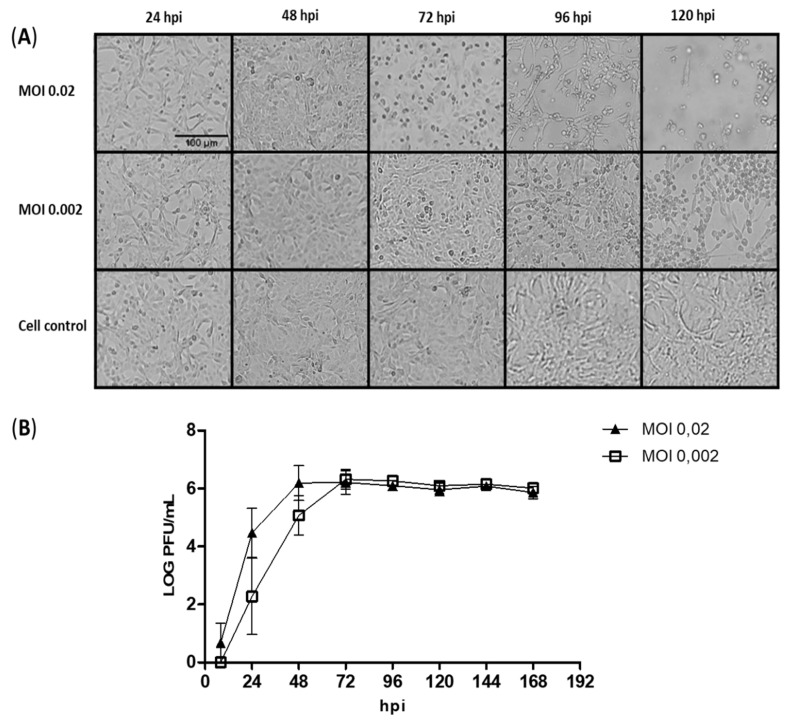
Viral kinetics of ES 2916/2015 Zika virus (ZIKV) in VERO cells at MOI 0.02 and MOI 0.002. (**A**) Cell monolayer images demonstrating the cytopathic effect of ZIKV over time. (**B**) Viral titers determined by plaque-forming unit (PFU) assay.

**Figure 2 viruses-14-01867-f002:**
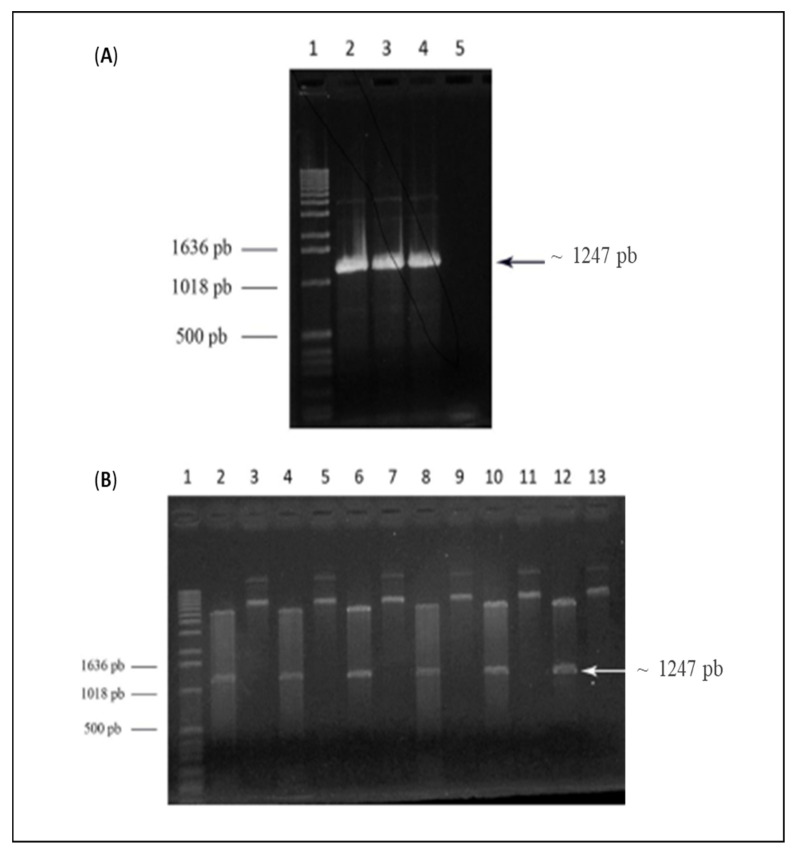
One percent agarose gel. (**A**) Electrophoresis of the PCR product of the Zika virus recombinant protein E (E-ZIKVre) gene in the commercial vector pUCIDT + E-ZIKVre. Lane 1—Molecular marker 1kb DNA Ladder™ (Life Technologies); line 2–4—PCR product E-ZIKVb; lane 5—negative control. (**B**) Electrophoresis of the enzymatic cleavage digestion products of the pET28a + E-ZIKVre construct with restriction endonucleases NcoI and HindIII. Lane 1—Marker 1kb DNA Ladder™ (Life Technologies); lane 2—digested clone 1; lane 3—undigested clone 1; lane 4—digested clone 2; lane 5—undigested clone 2; lane 6—digested clone 3; lane 7—undigested clone 3; lane 8—digested clone 4; lane 9—undigested clone 4; lane 10—digested clone 5; lane 11—undigested clone 5; lane 12—digested clone 6; and lane 13—undigested clone 6.

**Figure 3 viruses-14-01867-f003:**
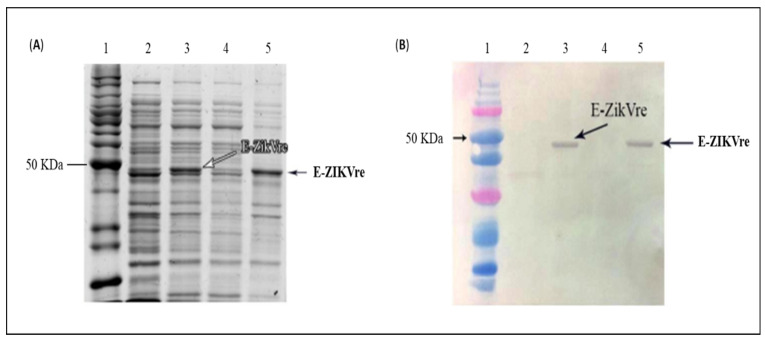
Evaluation of E-ZIKVre expression and solubility analysis in *E. coli SHuffle*. (**A**) SDS-PAGE gel. Line 1—benchmark; line 2—non-induced culture; line 3—induced culture; line 4—soluble fraction; line 5—insoluble fraction. (**B**) Immunoblot assay. Line 1—dual color; line 2—non-induced culture; line 3—induced culture; line 4—soluble fraction; line 5—insoluble fraction. Primary antibody: Anti-His.

**Figure 4 viruses-14-01867-f004:**
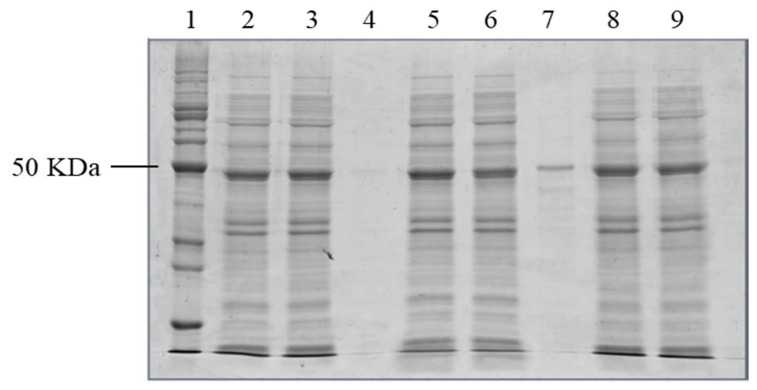
SDS-PAGE of the E-ZIKVre protein solubilization assay expressed in *E. coli SHuffle*. Line 1—benchmark; line 2—crude extract 4 M urea; line 3—soluble fraction 4 M urea; line 4—insoluble fraction 4M urea; line 5—crude extract 5 M urea; line 6—soluble fraction 5 M urea; line 7—insoluble fraction 5 M urea; line 8—crude extract 6 M urea; line 9—soluble fraction 6 M urea; line 10—insoluble fraction 6 M urea.

**Figure 5 viruses-14-01867-f005:**
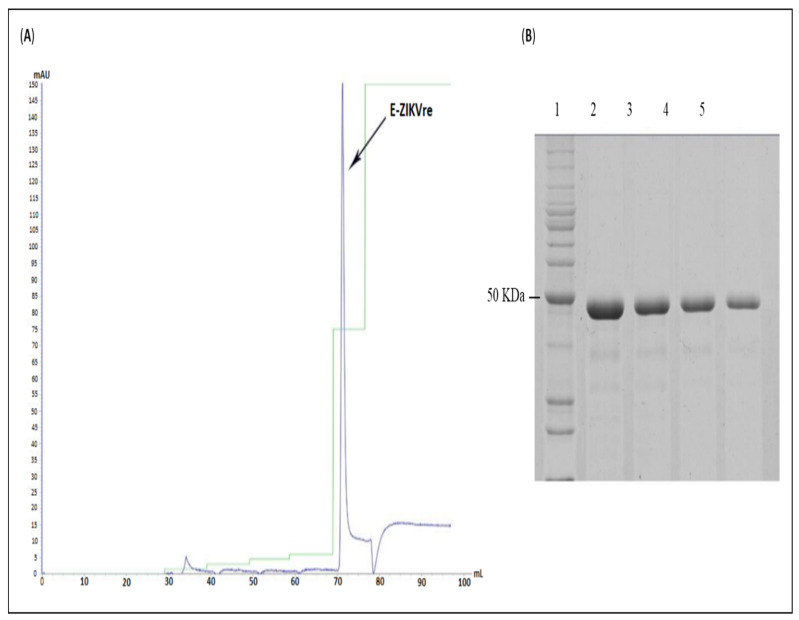
Analysis of the E-ZIKVre protein purification step. (**A**) Chromatogram of purification of protein by IMAC on Ni^2+^ column. On the vertical axis are the absorbance values in mAU (l = 280 nm) and on the horizontal axis are the volumes in milliliters. The blue peak represents the E-ZIKVre protein. Elution in 20 mM Tris-HCl + 1 mM EDTA + 0.5% triton x-100 + 4M urea (pH 8.0). (**B**) SDS-PAGE of E-ZIKVre protein purification product by IMAC. Line 1—benchmark; lines 2–5—purified fractions of E-ZIKVre protein.

**Figure 6 viruses-14-01867-f006:**
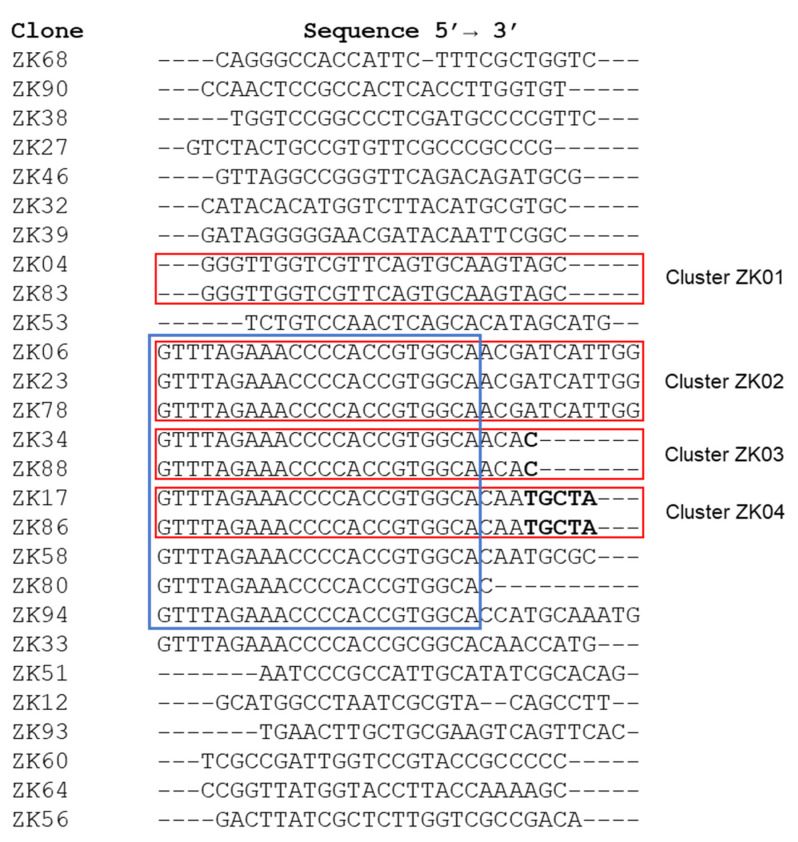
Alignment of the sequences obtained after selection against the whole Zika virus particle. Red rectangles indicate four clusters of aptamers. Blue square shows nucleotide homology between sequences. The nucleotides in bold were removed.

**Figure 7 viruses-14-01867-f007:**
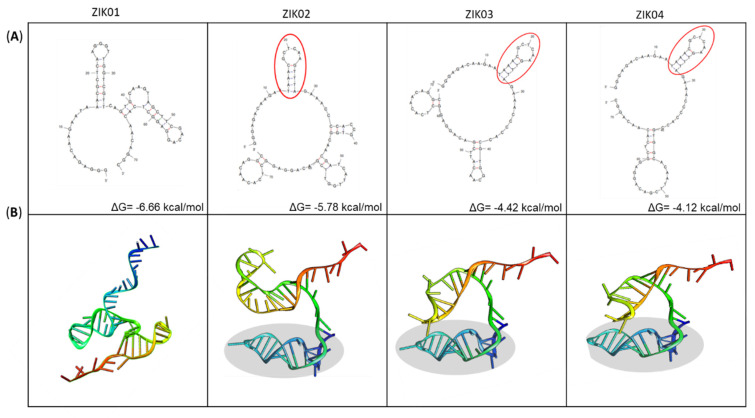
Prediction of secondary and tertiary structures of the four selected aptamers for Zika virus. (**A**) Secondary structure of the aptamers by Mfold web server. Parameters: linear DNA; folding temperature = 25 °C; [Na+] = 0.1; [Mg++] = 0.01; unit = M; ΔG = free energy; the circled areas show the same hairpins structures in 3 aptamers. (**B**) Computational prediction of the 3D structures of the aptamers using Chimera program. The highlighted areas indicate the same tertiary structure in three aptamers.

**Figure 8 viruses-14-01867-f008:**
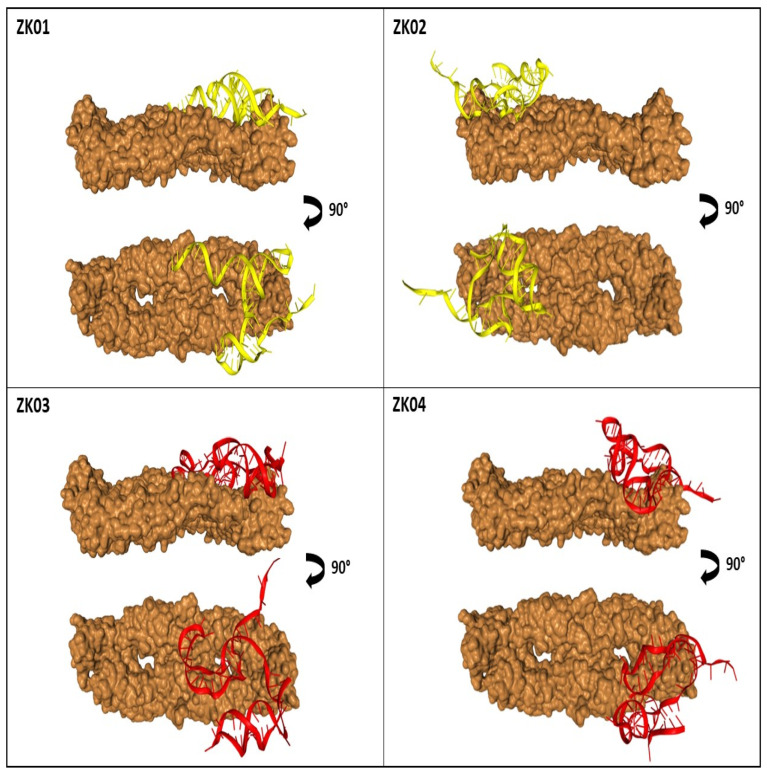
Molecular dynamics simulation of the selected aptamers and dimeric ZIKV E protein by the HDOCK server. Protein E is represented at two angles: the first, showing the aptamer binding on the outside of the protein; the second, with a 90° rotation, detailing the exact location of interaction.

**Figure 9 viruses-14-01867-f009:**
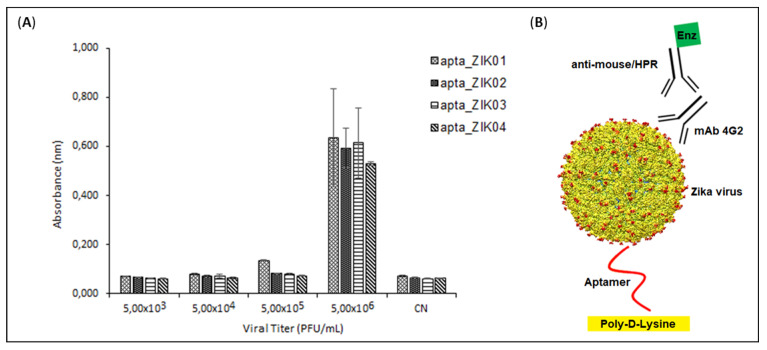
In-house ELISA: (**A**) Reactivity of the four aptamer clusters with the ES 2916/2015 Zika virus and negative control (no virus); and (**B**) Schematic representation of the test.

**Figure 10 viruses-14-01867-f010:**
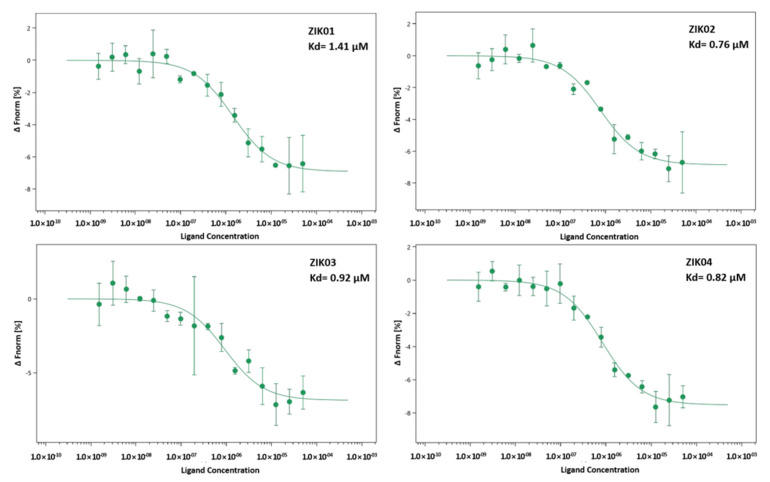
Thermophoretic assay to compare affinity between aptamers and E-ZIKVre protein. MST parameters: target concentration: 20 nM; ligand concentration: 50 µM–0.00153 µM; excitation power: 76%; temperature: 25 °C.

**Figure 11 viruses-14-01867-f011:**
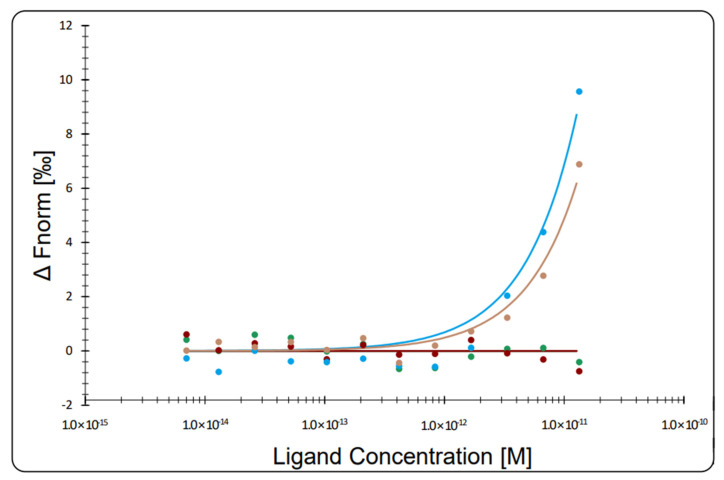
Thermophoretic assay to compare affinity between aptamers and Zika virus (ZIKV) and Dengue virus (DENGV) particles. The figure shows the plot of baseline-corrected normalized fluorescence (ΔFnorm) (‰) versus ligand concentration, in a heating time of five seconds.

**Table 1 viruses-14-01867-t001:** Docking position of aptamers in the E protein.

ZK01		ZK02		ZK03		ZK04	
aa	Position ^#^	aa	Position ^#^	aa	Position ^#^	aa	Position ^#^
SER	64A	MET	68A	ARG	2A	ER	306A
SER	66A	ALA	69A	VAL	46A	THR	309A
GLN	89A	SER	70A	THR	47A	ALA	310A
LYS	118A	ASP	71A	GLU	62A	ALA	311A
PHE	119A	SER	72A	SER	64A	THR	313A
ALA	120A	ARG	73A	SER	122A	PHE	314A
CYS	121A	TYR	81A	LYS	123A	THR	315A
SER	122A	ASP	83A	LYS	124A	ILE	317A
LYS	123A	GLY	102A	ARG	138A	GLN	331A
LYS	124A	ASN	103A	MET	140A	TYR	332A
GLY	145A	HIS	249A	GLU	162A	ALA	333A
SER	146A	ALA	250A	LYS	166A	GLY	334A
GLU	162A	LYS	251A	ASN	207A	THR	335A
ASN	163A	ARG	252A	GLU	262A	ASP	336A
ARG	164A	ARG	2B	ARG	283A	GLU	367A
GLY	182A	HIS	27B	THR	313A	SER	368A
ASP	230A	VAL	46B	PHE	314A	THR	369A
THR	231A	THR	47B	THR	315A	GLU	370A
GLY	232A	MET	140B	GLN	331A	ASN	371A
THR	233A	SER	146B	GLU	367A	SER	70B
PRO	234A	GLU	162B	GLU	370A	ASP	71B
TRP	236A	ASN	163B	ASN	371A	SER	72B
ASP	278A	ARG	164B	LYS	373A	ARG	73B
GLY	279A	LYS	166B	GLU	62B	CYS	74B
ARG	283A	ASP	278B	ALA	63B	THR	76B
LYS	301A	ARG	283B	SER	64B	GLN	77B
THR	309A	LYS	301B	ILE	65B	GLY	78B
ALA	310A	VAL	303B	SER	66B	GLU	79B
ALA	311A	TYR	305B	ASP	67B	TYR	81B
THR	313A	GLN	331B	MET	68B	LEU	82B
PHE	314A	THR	335B	GLN	77B	ASP	83B
THR	315A	PRO	338B	TYR	90B	GLY	104B
ILE	317A	THR	366B	TRP	101B		
GLN	331A	GLU	367B	GLY	102B		
ALA	333A	SER	368B	ASN	103B		
GLY	334A	THR	369B	LYS	118B		
THR	335A	GLU	370B	SER	122B		
VAL	364A	ASN	371B	LYS	251B		
THR	366A	SER	372B	ARG	252B		
GLU	367A	LYS	373B	THR	254B		
SER	368A			VAL	255B		
THR	369A			VAL	256B		
GLU	370A			VAL	257B		
ASN	371A			LEU	258B		
SER	372A			SER	260B		
LYS	373A						
MET	374A						
MET	375A						
LYS	394A						
ARG	73B						
CYS	74B						
GLN	77B						
TYR	81B						
ASP	83B						
GLY	104B						

*^#^ A = monomer A; B = monomer B.*

## Data Availability

Not applicable.

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
