# Peer review of "Selection and Characterization of Single-Stranded DNA Aptamers of Diagnostic Potential against the Whole Zika Virus"

_viruses, 2022, doi:10.3390/v14091867_

Round 1

Reviewer 1 Report

Excellent manuscript with only few minors comments. The Zika virus produces life long disabilities on unborn babies and it is essential for mothers-to-be to have access to a fast diagnostic method. This aptamer could be used as such diagnostic tool and help to avoid the spread of this infection, thus avoding the contact between ill and non-ill patients.

from abstract: "and accurate diagnosis crucial" add "are crucial".

line 81, "zika" should be "Zika".

line 246, no space between 37 and the unit. Please check the spaces in between numbers and the units, in some cases there is space and in some there is no space.

figure 2, please check the font, it is not uniform.

line 439, replace "e" with "and".

Author Response

All points requested by the referee have been apropriately addressed. Namely:

In the abstract: we have added "is crucial".

line 81, "zika" has been replaced by "Zika".

line 246 and in all the Materials and Methods section, and remaining document we have checked and added spaces between numbers and their respective units

figure 2, the font has now been made uniform in the figure legent.

line 439, the "e" has been substituted by "and".

We thank for the constructive comments and corrections, which have now all been addressed acordingly. 

Reviewer 2 Report

There appear to be some references on Zika virus aptamers that are missing such as Lee and Zeng, Analytical Chem. 2017 Dec 5;89(23):12743-12748 and a few others regarding aptamers against the NS1 protein of Zika.

This reviewer has a serious concern about the use of poly-lysine to coat the ELISA (really ELASA) plate wells as shown in Figure 9B.  Firstly, why not coat the microwells with the virus or viral proteins using NaHCO3 at pH > 8 to screen the candidate aptamers as the authors did for the SELEX development?  The problem with coating the wells with poly-lysine is that poly-lysine does not create a covalent immobilization bond with the aptamers.  Poly-lysine is clearly cationic and electrostatically interacts with the anionic phosphate backbone of the aptamers. I know that Stearns et al. (ref. 43) claims to have used this in 2016 for immobilizing antibodies, but the aptamers might desorb from the poly-lysine-coated plate when they encounter their cognate target (the Zika virus).  This occurs with graphene oxide substrates that aptamers adsorb to and desorb in the presence of their targets.  So, if any aptamers desorb from the scheme in Figure 9B, they will not even be detected!  They would get washed away in the ELISA/ELASA process before colorimetric detection.  And if the poly-lysine is strong enough to hold the aptamers down, it may tend to linearize the aptamer structure by flattening the phosphate backbone and altering aptamer 3D structure and binding to the virus. I'm not sure there is a way to logically  defend what the authors have done here, but they at least need to address it better.  It may be a major flaw in the paper.  I'd like to see them repeat the ELASA/ELISA by immobilizing the viruses (serial 10-fold dilutions) and adding biotinylated candidate aptamers plus streptavidin-peroxidase.  The ELASA/ELISA results could be radically different then. 

Can the authors legitimately call their aptamers "high affinity" reagents in line 511 when Figure 10 states microMolar Kd values by thermophoretic assay?  

The axes of Figures 5 and 10 are very hard to read.  Can the authors at least increase contrast or sharpness to make the axes more legible?

Minor points:

Line 216, should probably change to read Applied Biosystems instead of just Applied.

Lines 226, 322 and maybe elsewhere, there is random capital bold text, please fix that throughout.  

Author Response

We have now tried to respond to all the referees comments and address any concerns. Our response is presented in the attached document. 

Reviewer 3 Report

The manuscript titled "Selection and characterization of single stranded DNA Aptamers of diagnostic potential against the whole Zika virus" presents the selection of specific aptamers for whole Zika virus. The authors have produced extensive work to cover the whole story however there are a few questions that needs to be answered.

1. The authors used whole Zika virus particles for the selection and the characterization part however for the molecular docking part they used the E-ZIKVre protein. What exactly prompted the authors to select this protein as being the target of the selected aptamers? If the authors intend to justify the molecular docking simulation studies then additional results need to be shown to prove that the aptamers target the E-ZIKVre protein on the virus particles.

2. Line number 192: Authors should mention the concentration of library used.

3. Line number 194: it should be unbound nucleotide sequences.

4. Line 200: The amplification condition mentions that there were 100 cycles of PCR which is insanely high for aptamer selection. Usually the PCR amplification cycles are never more than 20 to avoid sequences that are easily amplified from taking over the unique sequences. Authors should look into if that was a typo.

5. The ELISA experiment should include a negative control for comparison, where the virus is absent but everything else is added.

6. The Kd value for ZK01 does not match with the figure value.

Author Response

In order to address the referees comments, some modifications have been made in the text of the article, as shown below:

1) It is true that having made the selection of the aptamers against the whole virus, there was inicially no way of knowing whether these were indeed selected against the E protein. However, we tested the aptamers against the whole virus and the E protein, and we found that the aptamers bound to both, justifying the suspission that this would be the target, based on the abundance of the protein on the viral surface. Thus, in the begining of the molecular docking section, we have added the following text to explain why the modeling has been made using the E protein and the aptamer: 

"Protein E is a glycoprotein that involves the entire surface of flavivirus viral particles. Each glycoprotein E monomer forms stable homodimers, and 90 dimers assemble to form the outer envelope of the infectious virus. Protein E is the main target of neutralizing antibodies against the Zika virus and the likely target of the selected aptamers. We verified this assumption using recombinant protein E in thermophoretic analyses, to confirm that the aptamers bind indeed to this protein, verify the binding profile and calculate dissociation constants. We also performed the assays with the intact virus to confirm the aptamer-target specificity, using a Dengue virus pool as a control (see next section).

 Having confirmed the interaction of the selected aptamers with the E protein, molecular dynamics simulation between aptamers and ZIKV E protein was predicted using the HDOCK server. "

2) The concentration of the library has been added in line 192, as per request of the referee. 

3) Text has been modified in line 194 to read "unbound nucleotide sequences"

4) The amplification has indeed been 100 cycles, but this is not a tradicional PCR (where it is common to use 20 or 30 cycles), but a unidirectional PCR, that only makes one copy per cycle, with suficient reverse primer to amplify once (or a few times) the template to double stranded, and 100fold more forward primer, so that it only copies 100 times in one direction. This is based on a protocol used successfuly by our group on various occassions and previously published, as in the reference mentioned in the article (Simmons et al, 2012; reference 41 in the article). To avoid this confusion, we have added in the text the term "unidirectional PCR" on line 197. 

5) We apologise for the lack of the representation of the negative control in the figure. This has now been added on figure 9 as suggested by the referee, and the appropriate modification in the figure legend to include this information.

6) The value of the Kd is correct, but the unit was wrong in the text. It has now been modified to read nM instead of uM. Thus, 1,414 nM is igual to 1.41uM shown in the figure. 

English revision has been made to further improve quality of the text

Round 2

Reviewer 2 Report

Edits appear adequate.